# A deep learning-based approach to enhance accuracy and feasibility of long-term high-resolution manometry examinations

Alexander Geiger [1] ✉, Lars Wagner [1], Daniel Rueckert[2,3], Dirk Wilhelm[1,4] & Alissa Jell [1,4]

## Abstract

**Background** High-resolution manometry (HRM) is the gold standard for diagnosing esophageal motility disorders. However, its short-term laboratory setting often fails to capture intermittent abnormalities. Long-term HRM (LTHRM, up to 24h) provides richer insights into swallowing behavior, but the resulting data volume is immense. Manual analysis by medical experts is laborious, time-consuming, and prone to errors, limiting its clinical feasibility.

**Methods** We propose a deep learning-based approach for automatic analysis of LTHRM data. Our method detects both swallow events and secondary non-deglutitive motility disorders with high accuracy. Detected swallows are then clustered into distinct classes of similar events, creating a structured overview of motility patterns and their frequency. This reduces the analytical burden by allowing clinicians to focus on a small number of representative swallows rather than manually reviewing thousands of individual events. We evaluate our pipeline on 25 LTHRMs that were meticulously annotated, resulting in a dataset of more than 23,000 expert-labeled events.

**Results** Our approach is able to detect more than 94% of all relevant events in LTHRM sequences, while the subsequent clustering is able to capture and group all relevant events into distinct swallow groups. To evaluate the overall approach, we conduct a user study with medical experts, demonstrating its effectiveness and positive clinical impact.

**Conclusions** Our findings demonstrate that deep learning-based approaches to analyze LTHRM examinations are capable of providing a more reliable and efficient diagnostic process, ultimately making LTHRM assessments more feasible in clinical care.

## Plain language summary

High-resolution manometry (HRM) is the standard test for diagnosing swallowing and esophageal motility disorders by analyzing swallowing and pressure patterns in a patient. This method uses short recording time and can often miss intermittent problems. Long-term HRM (LTHRM) captures a fuller picture but produces large datasets that are difficult to analyze manually. We present a computer learning-based method that automatically detects swallow events and abnormal motility patterns in these larger datasets and clusters them into groups of similar events. This structured representation reduces the data burden and allows experts to focus on representative examples. Evaluated on 25 recordings with over 23,000 labeled events, our method is able to detect more than 94% of all relevant events and is shown in a user study to improve efficiency in clinical interpretation.

Benign esophageal diseases present significant health and socio-economic challenges, especially for aging populations. Dysphagia, characterized by difficulties in swallowing food, drinks, or even saliva, becomes increasingly prevalent with age, posing challenges for patients and healthcare providers[1–3]. While conditions such as gastroesophageal reflux disease have garnered considerable attention and research, swallowing disorders like dysphagia, which may arise from impairments in the oral, pharyngeal, or esophageal phases of swallowing, remain less understood and frequently overlooked[4,5]. Patients with sporadic esophageal disorders often experience severe symptoms, yet diagnosing these intermittent conditions can be immensely challenging. High-resolution manometry (HRM) is the gold standard for diagnosing esophageal motility disorders[6–8]. Conventional HRM is typically conducted in controlled laboratory settings, currently following a standardized swallow protocol according to the Chicago

[1]Research Group MITI, TUM University Hospital, School of Medicine and Health, Technical University of Munich, Munich, Germany. [2]Chair for AI in Healthcare and Medicine, Technical University of Munich (TUM) and TUM University Hospital, Munich, Germany. [3]Department of Computing, Imperial College London, London, United Kingdom. [4]Department of Surgery, TUM University Hospital, School of Medicine and Health, Technical University of Munich, Munich, Germany. ✉e-mail: alexander.geiger@tum.de

Classification 4.0 (CC4.0)[9] within a limited timeframe. HRM aims to analyze swallow motility patterns by measuring the pressure within the esophagus of a patient[6]. A manometry catheter with multiple pressure sensors is inserted transnasally into the esophagus and collects the pressure information along the tube. In short-term settings, the patient is given predefined 5-ml wet boli to swallow. Afterwards, the collected data is analyzed by medical experts. This approach is already proven to miss intermittent dysfunctions that manifest outside of the testing period[10,11]. To address these limitations, HRM monitoring can also be extended to a long-term setting (up to 24 h)[12], where the patient is spending a regular day with the catheter inserted. They are asked to maintain their usual daily routines and activities in order to maximize the representativeness of the collected data. Afterwards, the data is analyzed by a medical expert. Figure 1 shows the general workflow of LTHRM examinations.

However, this expansion presents new challenges, as the increased data volume necessitates sophisticated, manual, and time-consuming methods for analysis and interpretation[13,14]. Each examination generates a large volume of complex data, which is usually represented as color-coded spatio-temporal pressure topography plots. Medical professionals must meticulously review these plots in order to identify and analyze all swallow events. The total number of swallow events often exceeds 1000 swallows per patient. Therefore, the data analysis process is not only labor-intensive but also time-consuming, often taking up to 3 days per patient, depending on the complexity of the motility patterns observed. The need for such detailed analysis can limit the throughput of HRM studies in a clinical setting, potentially creating bottlenecks in patient care and delaying diagnosis and treatment.

Another major challenge is the high level of specialized domain knowledge required to analyze LTHRM data. Medical professionals must be experienced at distinguishing between normal and pathological motility patterns, which involves understanding the nuances of esophageal physiology and the various esophageal motility disorders, such as achalasia, diffuse esophageal spasm, ineffective motility and esophagogastric junction disorders. Additionally, interpreters must be familiar with the Chicago Classification. Mastering this system requires significant training and experience, underscoring the importance of specialized expertise in ensuring accurate and clinically meaningful interpretations of HRM data.

A lot of research has been done on using computer-aided methods, including artificial intelligence (AI), in the diagnosis of esophageal disorders[15,16]. One specific area is the analysis of HRM data, which includes a wide range of tasks such as automated sphincter motility analysis[17,18] or probe position failure detection[19], as well as the intuitive visualization of HRM data[20,21]. When it comes to methods aiming to support the medical assessment of swallow events, the automatic interpretation of HRM data has been an active area of research. The standardized HRM examination necessitates the manual evaluation of up to 21 swallow events per patient, depending on the underlying pathology[9]. However, it is widely acknowledged that there is limited inter-rater reliability due to varying expertise levels in the evaluation of HRMs[22,23]. Consequently, various methodologies have been developed to aid and standardize this process, aiming for automatic categorization of these swallowing events according to predefined classification standards. Hoffman et al.[24] used an artificial neural network to classify swallows as safe, penetration, or aspiration based on HRM and impedance. In another work, Hoffman et al.[25] used a neural network-based approach to identify patterns of disordered swallowing in HRM plots.

Carniel et al.[26] developed a physiological parametric model for each swallow category, which Frigo et al.[27] then subsequently used to compare new swallows to these models in order to identify the closest category. Additionally, manometry and impedance recordings can be used to gain information about the distension and contraction of the esophagus, which can be used as features to classify motility disorders[28,29]. Popa et al.[30] categorized manometry images into ten distinct clusters utilizing a convolutional neural network (CNN) model. In another study, the classification was refined by combining two models, one for identifying the integrated relaxation pressure and one for identifying the swallowing disorder, in order to automate the Chicago Classification algorithm[31]. Kou et al.[32] used a long short-term memory (LSTM)-based approach in order to classify swallows into predefined categories. Additionally, Kou et al.[33] used a multi-stage ML-based classification approach on standardized HRMs, first classifying three swallow parameters, before having another mixture of models to predict the final diagnosis. Wang et al.[34] used a multi-stage approach as well, first identifying the regions of the swallows in the manometry inputs, before classifying these motility sequences using a temporal model based on a CNN and bidirectional convolutional LSTM architecture, allowing to classify the sequences into one of three classes. Furthermore, in order to better understand swallow-level data, Kou et al.[35] developed a generative model based on a variational auto-encoder that is able to generate swallow manometries depicting several physiologically-meaningful motility patterns.

These studies have demonstrated robust methodologies for automated swallow classification, achieving accuracies from 81% to 91% when classifying the swallows into predefined disorder classes[30–34]. Notably, these classifications are based on manually extracted individual swallow events that are separated by clearly defined intervals of at least 30 seconds in short-term HRM. LTHRM goes beyond these limited laboratory studies, leading to interlocking swallow events and a total of 900 to 1500 swallows within 24 hours, thus making manual annotation impractical and therefore showing the need to have an automated swallow detection method in LTHRM.

In previous research work[13,14], a first automated detection algorithm was introduced, which is able to identify swallows by spotting upper esophageal sphincter (UES) relaxations in the proximal five channels without the need of previously manual annotations of the anatomical landmarks by medical experts. However, this limits the evaluable HRMs to those patients with the upper esophageal sphincter well visible in the predefined uppermost five channels. The method achieves a reported recall of 89.7%.

In this work, we extend the existing approach by employing a deep learning (DL)-based method that enables more robust swallow event detection independently of any anatomical landmarks. Unlike previous methods that rely solely on sensors positioned within the manually defined upper esophageal sphincter region, our approach utilizes data from all sensors, thereby increasing the robustness of swallow detection. Building on this, we then propose an automated clustering system designed to streamline the analysis of long-term HRM (LTHRM) recordings, aiming to expedite the diagnostic process and alleviate the burden on healthcare professionals. Specifically, we develop a DL-based procedure leveraging CNNs to capture the two-dimensional characteristics of swallow manometries. We thoroughly evaluate the swallow detection performance on a total of 25 LTHRMs comprising over 23,000 swallow events, achieving a 94% average recall-score with our method, which outperforms both a non-

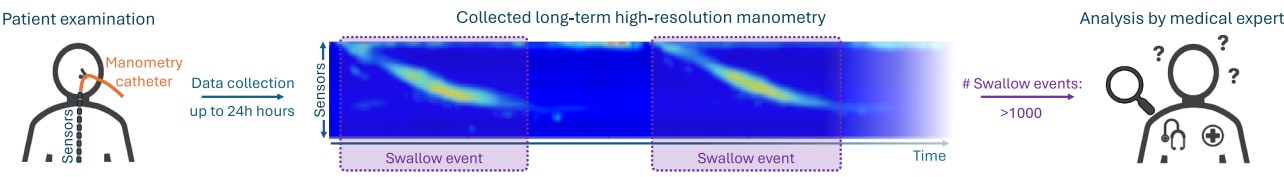

**Fig. 1 | General workflow of current LTHRM examinations.** The manometry data is collected from a high-resolution manometry catheter, containing 36 pressure sensors, that is inserted into the esophagus of a patient. The data is collected for up to 24 hours. The resulting manometry data typically contains more than 1000 swallow events. The full sequence has to be analyzed by medical experts in order to identify all swallow events in the data and analyze the specific details of each swallow event.

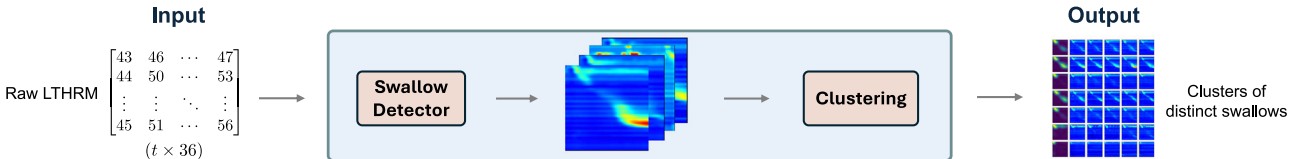

**Fig. 2 | Overview of the computational pipeline.** The pipeline consists of a swallow detector and a subsequent clustering of the detected swallows.

machine learning (ML) baseline and a commercial LTHRM evaluation tool. Our subsequent clustering approach categorizes detected swallows into distinct classes, grouping similar events together and thereby reducing the manual evaluation time of relevant motility patterns for clinicians. Finally, we provide an evaluation study of our approach in terms of its clinical value by experienced clinicians, demonstrating that it is not only less time-consuming but also has the potential to be less stressful for medical experts, without impairing evaluation quality compared to the manual approach.

## Methods
Our proposed computational pipeline employs a two-step approach. The first step involves automated detection of all swallowing events in the collected manometry data. Similar swallows are subsequently clustered into groups in the second step, resulting in the presentation of only a few representative images from each swallow cluster to medical experts for further diagnosis. The overall approach is depicted in Fig. 2.

### Data set
The data set consists of 25 LTHRMs of patients with suspected intermittently occurring motility disorders of the esophagus, collected at the TUM University Hospital Rechts der Isar in Munich, Germany. The patient data was collected and used with ethical approval and informed consent. Approval was granted by the Ethics Committee of the TUM University Hospital rechts der Isar (No. 135/19S). All patients underwent endoscopy prior to HRM to rule out malignancy and other structural causes of dysphagia. After a fasting period of at least 6 hours, the manometry catheter was placed transnasally, and a standardized examination was performed according to CC4.0 protocol. In the absence of functional reasons for dysphagia, the patients were introduced to the specifics of the extended LTHRM examination. They were advised to keep a recording of their meals, body position, symptoms and maintain their daily routine as much as possible to obtain representative and traceable measurements. All measurements, irrespective of activity (including mealtime segments), were retained for subsequent data analysis, as their inclusion is essential for capturing realistic circadian and behavioral patterns.

The HRM catheter contains 36 circumferential pressure sensors located 1 cm apart. Manometric values are collected at a sampling rate of 50 Hz, resulting in a manometry matrix denoted $\mathbf{M} \in \mathbb{R}^{36 \times t}$, where $t$ is the number of measurements. The values are smoothed by using a moving average window across the $t$ dimension to get $\hat{\mathbf{M}}$, where $\hat{\mathbf{M}}_{i,j} = \frac{1}{w}\sum_{k=j}^{j+w-1} \mathbf{M}_{i,k}$, with $i = 1\dots36$; $j = 1\dots t - w + 1$, and $w = 30$ in our experiments. The values in $\hat{\mathbf{M}}$ are then clipped (to achieve values in a default range between $-200$ and $300$ mmHg, thus removing extreme outliers that fall outside the physiologically valid pressure range) and rescaled to a standardized range between 0 and 255 (to normalize the data while preserving the critical spatio-temporal patterns). After collection, all swallow event starts were first annotated by a single experienced medical expert, where the swallow start was defined by the relaxation of the UES, which reliably marks the initiation of a swallow event. These initial annotations were then independently reviewed and verified by a second expert to ensure consistency and accuracy. This resulted in a total of over 23,000 labeled swallow events.

### Swallow detection
The initial step involves the automated detection of swallow events in LTHRM. For this purpose, we design and implement two approaches that

are described below. One is serving as a non-ML baseline, while the second is our proposed DL-based detection method. In the evaluation, the detection performance of the approaches is evaluated and compared to a third method, which is the (to our knowledge) only commercially available LTHRM evaluation software (ViMeDat v5.1.6.0, Standard Instruments, Germany).

**Non-ML baseline: threshold-based approach.** As the pressure along the 36 sensors behaves characteristically during swallowing (e.g., pressure increase in the pharynx/esophagus), we use a threshold-based approach to find peaks in the pressure values to identify swallow events. This serves as a non-ML baseline. The method takes the complete preprocessed manometry $\hat{\mathbf{M}}$ as an input. This matrix is then converted to a binary matrix $\mathbf{M_b}$ and a moving sum is applied to highlight regions with a series of consecutive ones before summing across all sensors. Formally,

$$\mathbf{M}_{\mathbf{b}i,j} = \begin{cases} 1, & \text{if } \hat{\mathbf{M}}_{i,j} > 80 \\ 0, & \text{otherwise} \end{cases}; \quad \mathbf{M}_{\mathbf{s}i,j} = \sum_{k=i}^{i+w-1} \mathbf{M}_{\mathbf{b}k,j}; \quad \mathbf{r}_j = \sum_{k=1}^{36} \mathbf{M}_{\mathbf{s}k,j} \quad (1)$$

where $i$, $j$ are the same as before and $w = 20$. The resulting vector $\mathbf{r} = [\mathbf{r}_1, \dots, \mathbf{r}_t] \in \mathbb{R}^{1 \times t}$ is smoothed again to get the final vector $\hat{\mathbf{r}}$, such that $\hat{\mathbf{r}}_j = \frac{1}{w}\sum_{k=j}^{j+w-1} \mathbf{r}_k$ for $j = 1\dots t - w + 1$ and $w = 100$. In this vector, a peak finding algorithm detects all peaks above a certain threshold as swallows.

**DL-based approach.** We use a supervised learning approach to classify an input manometry sequence $\mathbf{I} \in \mathbb{R}^{36 \times 500}$ into one of two classes, where 1 resembles a swallow sequence and 0 a non-swallow sequence. We created a training set by automatically iterating over the annotated manometries $\hat{\mathbf{M}}$ and storing a window of length 500 beginning from each annotated swallow start (i.e., swallow windows are aligned such that the UES relaxation corresponds to the left edge of the swallow window). This results in a swallow tensor $\mathbf{S} \in \mathbb{R}^{s \times 36 \times 500}$, with $s$ being the number of swallows in the data set $\hat{\mathbf{M}}$. For the non-swallow events $\mathbf{N}$, we extract windows of identical dimension from segments of $\hat{\mathbf{M}}$ that occur between annotated swallow events (provided that the inter-swallow interval is sufficiently large to accommodate such a window). Since all swallow events were explicitly annotated, these inter-swallow segments can be confidently assumed to be free of swallowing activity. This procedure ensures that only genuine non-swallow segments are sampled. Consequently, the automated data set construction process yields a curated collection of swallow and non-swallow events with approximately balanced class sizes. In order to get the required input dimensions for the specific models, each sequence $\mathbf{I}_i \in \{\mathbf{S}, \mathbf{N}\}$ is rescaled such that $\mathbf{I}_i^{36 \times 500} \rightarrow \mathbf{I}_i^{224 \times 224}$. For the swallow classifier, we implement several CNNs as backbones to compare their performance in our specified task, namely GoogLeNet[36], MobileNet[37], EfficientNet[38], and RegNet[39]. We argue that CNNs are well-suited for the presented case, especially due to their strength in detecting spatial hierarchies and local patterns, which are especially important in esophageal manometries, where slight changes in pressure or flow across the spatio-temporal dimensions can indicate different swallowing conditions or abnormalities. Furthermore, they inherently provide translation invariance and in our case also include a native temporal component due to the temporal dimension of the input sequences. The models were pretrained on the ImageNet data set[40]. The dimension of the final fully connected layer of each model was adapted to match our two binary output classes. The models were trained

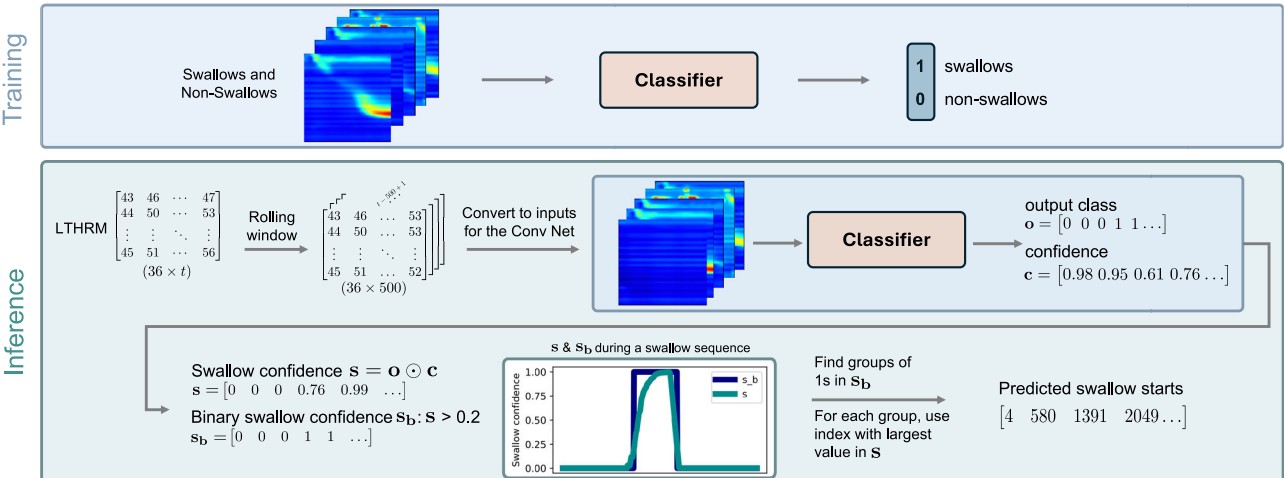

**Fig. 3 | Training and inference procedure of the swallow detection method.**
During the training stage, the classifier is trained to classify preprocessed manometry sequences into two categories (swallows and non-swallows). During inference, a rolling window is moved across the full LTHRM sequence. Each window is preprocessed similarly to the training sequences and then passed to the classifier for inference. The resulting list of output classes and confidences are then processed to identify all groups of swallows. For each swallow event, the predicted swallow start is returned.

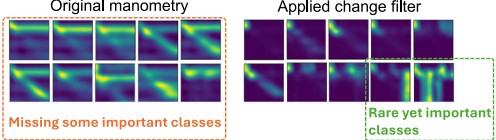

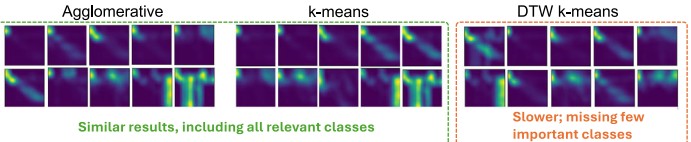

**Fig. 4 | Comparison of different clustering options. a** Shows the difference between using the pure manometry values and applying the change filter. Using the pure manometry values leads to missing a few rare, yet important classes, which can be seen when applying the change filter. **b** Shows the comparison of different clustering methods. Agglomerative clustering and k-means achieve similarly distinctive clusters that include all important classes, while DTW-based k-means results in slightly less distinctive clusters.

using the stochastic gradient descent optimizer with a learning rate of 3e-3 and a batch size of 128 for 20 epochs. The models were implemented in PyTorch and trained on a NVIDIA RTX A6000.

For the inference part, we perform the swallow detection by using a rolling window of length 500 over the test manometry $\hat{\mathbf{M}}$, resulting in $t - 500 + 1$ inference windows, denoted as $\mathbf{I}_i$ for $i = 0, \ldots, t - 500 + 1$, which are passed to the classifier model, such that for each $\mathbf{I}_i$ the output class and the corresponding confidence is computed. The resulting two vectors are the binary output class vector $\mathbf{o}$ and confidence vector $\mathbf{c}$, with $\mathbf{o}, \mathbf{c} \in \mathbb{R}^{1 \times (t - 500 + 1)}$. We then use the element-wise product $\mathbf{s} = \mathbf{o} \odot \mathbf{c}$ to obtain only the confidences of class 1 outputs. Afterwards, a moving averaging window is applied to smooth the values and convert the resulting vector $\hat{\mathbf{s}}$ to a binary vector $\mathbf{s_b}$, formally

$$\hat{\mathbf{s}} = [\hat{s}_1, \ldots, \hat{s}_j], \text{ with } \hat{s}_j = \frac{1}{w} \sum_{k=j}^{j+w-1} \mathbf{s}_k; \qquad \mathbf{s}_{\mathbf{b}i} = \begin{cases} 1, & \text{if } \hat{s}_i > 0.2 \\ 0, & \text{otherwise} \end{cases}$$

(2)

with $w = 20$. All groups of consecutive ones in $\mathbf{s_b}$ are considered a detected swallowing sequence. As we assume that the detection confidence is maximized when the swallow start aligns with the left edge of the analysis window, we use the maximum value in $\mathbf{s}$ as the predicted start of a swallow sequence. The full training and inference pipeline is depicted in Fig. 3.

**Clustering of similar events**

To cluster the detected swallows, we compared multiple clustering methods, namely standard k-means, agglomerative clustering, and dynamic time warping (DTW)-based k-means. In our evaluation, k-means and agglomerative clustering tended to result in very similar clusters, while the DTW-based k-means method took longer and produced slightly more homogeneous clusters, therefore favoring the first two methods. For the remainder of the evaluation, we therefore decided to focus on the agglomerative clustering. We also compared clustering the plain swallow images to clustering the images where a change filter is applied to the images beforehand, which specifically highlights pressure changes during a swallow. Based on a qualitative evaluation, we observed the best results using a change filter kernel $k$ of the form $k = [-1, 0, \ldots, 0, 1] \in \mathbb{R}^{1 \times 10}$, which is convoluted across manometry $\mathbf{I}_i$ for each swallow $i$. In essence, this filter highlights changes in the sensor values within a timeframe of ten measurements. The resulting change matrices $\mathbf{C}_i$ are squared and then rescaled such that $\mathbf{C}_i^{36 \times 500} \rightarrow \mathbf{C}_i^{50 \times 50}$, before applying a Gaussian filter for a final smoothing. Figure 4 shows a qualitative comparison of the different clustering methods, as well as the effect of using the pure manometry values compared to applying the specified change filter.

Prior to clustering, the matrices are flattened and a principal component analysis (PCA)[41] is performed to reduce the dimensions such that $\mathbf{C}_i^{50 \times 50} \rightarrow \mathbf{c}_i^{2500 \times 1} \rightarrow \text{PCA} \rightarrow \mathbf{c}_i^{30 \times 1}$. This results in the final vectors $\mathbf{c}_i$ for each detected swallow $i$, which are then passed to the clustering algorithm. The initial number of clusters is determined by running the clustering multiple times with different cluster numbers (from four to ten clusters) and selecting the number with the lowest mean intra-cluster distance.

We then consider all clusters containing 15% or more of all samples to be the main categories of swallows. This threshold was defined in close collaboration with a medical expert to ensure that the criterion separates the main cases from the more special cases in a medically meaningful way. As clinicians are mainly interested in special cases when looking for intermittently occurring motility disorders, the remaining samples are clustered

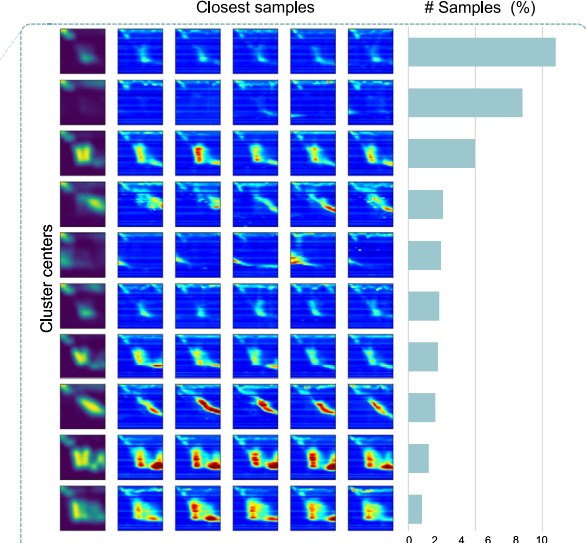

**Fig. 5 | Clustering of all detected swallows for a single patient. a** The swallows are first clustered into a certain number of classes, depending on the lowest mean intra-cluster distance. We consider all clusters containing 15% or more of all samples to be the main categories of swallows; the others are considered to be special classes. **b** As clinicians are mainly interested in special cases when looking for intermittently occurring motility disorders, the remaining samples are clustered a second time, in a more granular fashion.

a second time. A predefined cluster number of 10 is utilized, effectively creating distinct separations among clusters without resulting in an excessive number of similar groupings. The clustering result for one of the patients can be seen in Fig. 5 as an example.

### Evaluation study

In order to measure the impact of our detection and clustering approach, we conduct an evaluation study that compares the classic way of analyzing LTHRM with our proposed method. Specifically, we compare the following two methods in our setup: *Method 1: Conventional* In the classic way of analyzing LTHRM, the participants use the ViMeDat software and analyze the collected manometry values manually. They have access to the full recording, meaning they can iterate over the swallows and analyze all of the events that they are interested in. *Method 2: AI-based* In our proposed way of analyzing LTHRM, the participants get only the clusters of the swallows that are detected by our DL-based method, with the five closest samples to each cluster center and the respective frequency of each cluster.

To measure if there is a difference in evaluation performance between the two methods, we use two LTHRM recordings from two different patients (Case 1 and Case 2). Each participant analyzes one case using the *Conventional* method and the other using the *AI-based* method. Which case is analyzed using which method is switched among the participants. For each analysis, we allow a maximum of 10 minutes, but the participants can finish earlier. After each analysis, the participants have to answer questions about the analyzed case. In the end, after the participants are done analyzing both cases and answering the questions, they are asked to fill out the NASA TLX[42] questionnaire in order to measure the perceived workload for the participants during the analysis of the two cases. Figure 6 shows the study design, case details, and questions.

Due to the specialized nature of HRM examinations, only a limited number of skilled participants ($n = 6$) were available for this evaluation study. The participants are highly experienced doctors and practitioners of HRMs. While HRM is the gold standard for diagnosing esophageal motility disorders, the implementation of LTHRM is constrained by significant demands on time, personnel, and expertise in routine clinical practice. Consequently, only three medical centers in Europe are currently performing LTHRM, which limits the availability of comprehensive knowledge in this area. These constraints rendered the inclusion of a larger cohort of

experts impractical for this study. While the small sample size poses limitations, we argue that it is representative of the niche user population this tool is designed to support.

### Calculation of metrics

Our DL-based approach and ViMeDat are designed to detect the swallowing start. Therefore, we count a correct detection (true positive TP) if a predicted swallow start is in the range $[y - \frac{1}{2}d, \ldots, y + \frac{1}{2}d]$ where $y$ is the true swallow start and $d$ denotes the size of the window that we allow the prediction to fall into. In contrast, the baseline approach aims to detect swallows during the actual swallow event (not the start), therefore we count a TP if a predicted swallow is in the range $[y, \ldots, y + d]$. Any predicted swallows that are not counted as TP are considered as wrongly predicted swallows (false positives FP), while all true swallows that are not counted as TP are considered as not correctly detected (false negatives FN).

### Statistics and reproducibility

The performance of the swallow detection model was assessed using the commonly used metrics precision, recall, and F1-score. Hyperparameter tuning of the model was conducted using grid search. The results are calculated using a randomly divided fivefold cross-validation approach, where each fold contains the LTHRM recordings of five patients. This way, each of the five models is trained on 20 patients and evaluated on the remaining five patients. The results are then aggregated across the five runs.

### Results

Since our approach consists of two parts, the swallow detection as well as the subsequent grouping of similar swallows, we evaluate both parts individually.

### Swallow detection

A comparative analysis is conducted between our developed DL-based method, as well as the non-ML baseline and ViMeDat. It is noteworthy that the detection within this tool is tied to the initial manual annotation of anatomical features (i.e., localization of sphincters) by medical experts, while both our proposed methods are fully automated. The evaluation aims to demonstrate the efficacy of our proposed method within the medical context. To measure the swallow detection performance of the three methods,

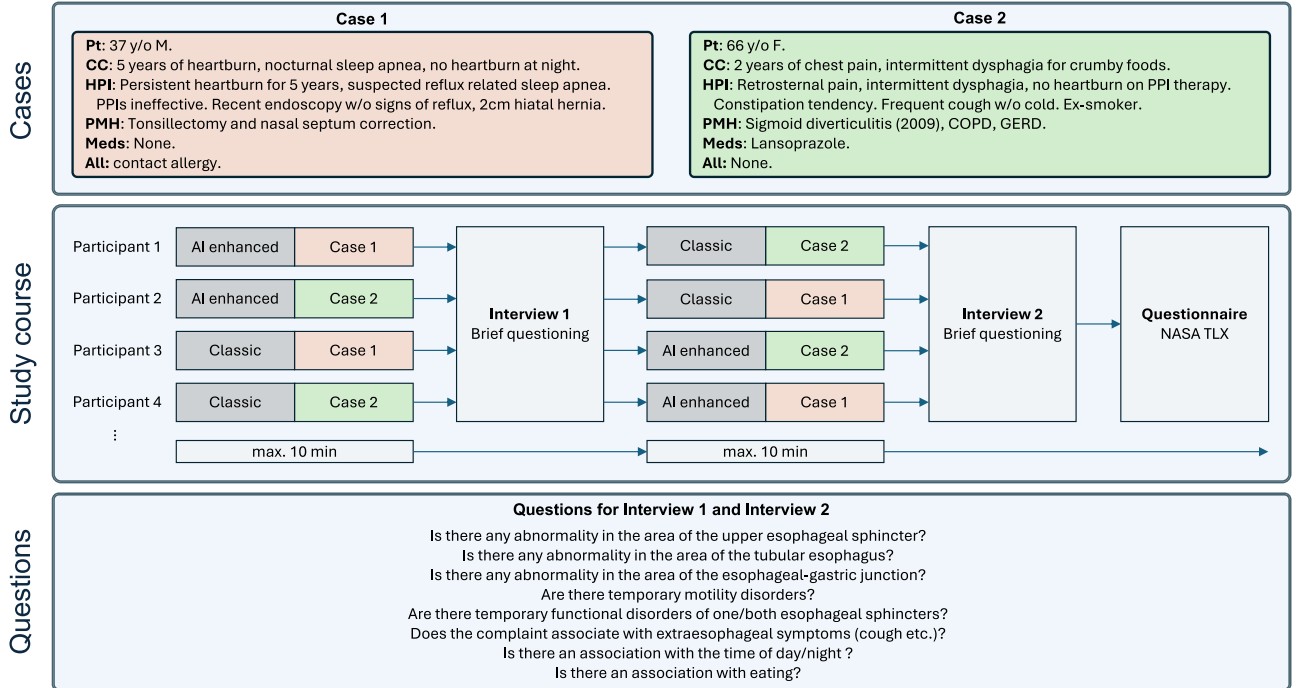

**Fig. 6 | Evaluation study design.** Two cases of LTHRMs were presented to medical professionals (top). To ensure comparability and potential order effects, a crossover study design was chosen (middle). After each timed case, the participants were asked brief questions on the cases (below), and their workload was assessed by the NASA TLX questionnaire.

**Table 1 | Comparison of the different detection methods**

| Method | Precision (%) | Recall (%) | F1-score (%) |
|---|---|---|---|
| Non-ML baseline | 29.58 ± 6.46 | 76.70 ± 9.77 | 38.70 ± 7.10 |
| ViMeDat | 85.73 ± 4.49 | 54.18 ± 15.86 | 61.59 ± 16.19 |
| Ours | **86.13 ± 2.98** | **94.07 ± 2.46** | **89.57 ± 2.59** |

We report the average metrics using a fivefold cross-validation along with their respective standard deviation ( ± ). Bold numbers indicate the highest performance in the respective category.

we deploy three different metrics suitable for the given classification task (i.e., precision, recall, and F1-score).

**Swallow detection results**. It can be observed that our proposed approach achieves the highest scores (see Table 1) in all metrics with a precision of 86.1%, recall of 94.1%, and F1-Score of 89.6%. This demonstrates that our presented algorithm consistently outperforms both the non-ML baseline as well as the commercially available ViMeDat tool. Additionally, Table 2 shows the performance metrics of the different lightweight backbone models trained for our DL-based approach, where it can be seen that MobileNet achieves the highest overall performance across precision, recall, and F1-score, while the other backbones perform slightly lower.

When investigating the outputs of all three methods, it can be observed that our approach is typically detecting the swallows slightly earlier than the labeled start, with the distance to the true start being centered around 35 measurements earlier. ViMeDat has a lower recall, meaning it detects less correct swallows, but those found are detected with lower average distance (centered around 0), indicating that the detected swallow starts are very accurate. The non-ML baseline is producing the swallow detections with the highest variance in the detection distances to the true start, arguably due to its design of not aiming to detect the swallow starts but those parts of the swallow that have the highest manometry values. In Supplementary Fig. 1,

we provide an overview of the distributions of distances for the predicted swallow start to the correct starts for the different methods.

Furthermore, in order to investigate how the window size $d$ influences the results, we performed the same experiments with two other values for $d$ (i.e., 400 and 800). In Table 3 we show the results for the three different values of $d$, demonstrating that the results obtained in all cases are consistent, independent of $d$.

**Clustering of similar events**

In order to provide an indication that the clustering approach is building meaningful clusters without losing important swallow events, we evaluated the most distant samples from their respective cluster center. We observed that for the majority of clusters, the most distant samples still objectively belong to this cluster, indicating that the produced cluster centers are indeed a representative and sufficient description of the different swallow types present in the LTHRM. From a medical perspective, it was also checked whether any of the most distant swallows would have actually had to be classified as a separate cluster due to its medical significance. In our evaluation, no such case was detected. In Fig. 7, the closest as well as most distant samples from each cluster center for two of the cases can be seen as an example. To additionally evaluate the clustering performance, a t-distributed stochastic neighbor embedding (t-SNE) algorithm[43] was applied to project the high-dimensional feature representations of all swallows into a two-dimensional space. The resulting visualization (see Fig. 7 for two examples), where the samples are colored according to their respective assigned cluster, demonstrates that samples assigned to different clusters by the agglomerative clustering algorithm are spatially separated and form distinct, cohesive groups. This separation between clusters suggests that the clustering approach effectively captures the underlying structure of the data and aligns similar samples into meaningful groups.

**Evaluation study**

The results of our evaluation study, which are visualized in Fig. 8, demonstrate that the proposed *AI-based* method offers notable improvements over

**Table 2 | Comparison of different model backbones for the classifier**

| Method | Precision (%) | Recall (%) | F1-score (%) |
|---|---|---|---|
| Classifier Backbone: MobileNet | **86.13 ± 2.98** | **94.07 ± 2.46** | 89.57 ± 2.59 |
| Classifier Backbone: GoogLeNet | 83.48 ± 4.39 | 94.05 ± 2.16 | 88.00 ± 3.23 |
| Classifier Backbone: EfficientNet | 83.86 ± 5.55 | 94.01 ± 1.80 | 88.23 ± 4.06 |
| Classifier Backbone: RegNet | 80.94 ± 4.59 | 91.06 ± 3.96 | 85.27 ± 4.25 |

We report the average metrics using a fivefold cross-validation along with their respective standard deviation (±). Bold numbers indicate the highest performance in the respective category.

**Table 3 | Comparison of the different detection methods, using different window sizes $d$**

| | Method | Precision (%) | Recall (%) | F1-score (%) |
|---|---|---|---|---|
| $d = 100$ | Non-ML baseline | n/a | n/a | n/a |
| | ViMeDat | **79.62 ± 5.12** | 50.28 ± 14.05 | 57.23 ± 14.36 |
| | Ours | 77.50 ± 7.63 | **84.68 ± 6.49** | **80.62 ± 7.18** |
| $d = 400$ | Non-ML baseline | 29.58 ± 6.46 | 76.70 ± 9.77 | 38.70 ± 7.10 |
| | ViMeDat | 85.73 ± 4.49 | 54.18 ± 15.86 | 61.59 ± 16.19 |
| | Ours | **86.13 ± 2.98** | **94.07 ± 2.46** | **89.57 ± 2.59** |
| $d = 800$ | Non-ML baseline | 37.76 ± 7.04 | 85.94 ± 9.56 | 48.22 ± 7.16 |
| | ViMeDat | 88.47 ± 3.86 | 58.46 ± 16.95 | 65.37 ± 16.66 |
| | Ours | **89.04 ± 1.92** | **97.65 ± 1.19** | **92.82 ± 1.36** |

Using $d \in [100, 400, 800]$, a correct swallow detection is counted for the MobileNet and ViMeDat approaches if the predicted swallow start is in a distance of at most $\pm \frac{d}{2}$ measurements from the true swallow start. Regarding the non-ML baseline, as it is not focusing on detecting the start of a swallow, but rather the actual swallow event, reducing the allowed distance to 100 is not applicable here, while for the other values of $d$, a correct swallow detection in this case is counted if the predicted swallow event is in range $[y + 200 - \frac{d}{2}, \ldots, y + 200 + \frac{d}{2}]$, with $y$ being the true swallow start. We report the average metrics over a fivefold cross-validation along with their respective standard deviation (±). Bold numbers indicate the highest performance in the respective category.

the *Conventional* method in several key aspects. First, the evaluation time was notably reduced for all participants when using the *AI-based* method compared to the *Conventional* method. Specifically, the mean evaluation time decreased from 488 to 179 seconds. This substantial reduction underscores the potential of the *AI-based* method to enhance efficiency, allowing the medical experts to complete LTHRM assessments in a fraction of the time required by the *Conventional* method. Second, the number of correct answers, which we use as a proxy for the achieved evaluation quality, remained consistent between the two methods, with participants achieving an average of 4.8 correct answers in both cases. The low absolute number of correct answers, coupled with the variability across the participants, high-lights the inherent difficulty of accurately analysing LTHRM data, even for trained professionals. This finding is consistent with existing literature, which has reported inter-observer variability in HRM assessments as a persistent challenge[22,23]. This strongly emphasizes the critical need for innovative examination tools capable of reducing subjectivity and improving the reliability of data analysis and interpretation. Third, participant confidence, rated on a scale from 0 to 2, was slightly higher for the *Conventional* method (average of 1.5) compared to the *AI-based* method (average of 0.8). This difference is likely attributable to the familiarity of the participants with the conventional workflow, which may have provided a higher sense of experience for them. While this result highlights a general potential barrier to the adoption of AI-based methods, it also underscores the importance of user training, as well as User Interface (UI) design, to increase confidence in unfamiliar novel tools. Lastly, results from the NASA TLX questionnaire indicated a slight decrease in perceived workload when using the *AI-based* method. The average total workload score decreased

from 26.7 to 26.0, with four of the six participants feeling a lower workload. This suggests that, for most users, our proposed approach simplifies the evaluation process of LTHRM.

In summary, the evaluation results highlight the potential of our proposed *AI-based* method to streamline LTHRM assessments by notably improving efficiency and reducing cognitive workload while maintaining comparable evaluation quality to the conventional approach. The lower confidence in the answers reported by participants suggests the need for further refinement of the user experience. Overall, our results underline the promise of AI-based approaches to standardize LTHRM assessments, reduce inter-observer variability, and ultimately improve the accuracy and consistency of data interpretation.

While the responsibility for ML-based diagnosis remains a critical aspect that requires careful monitoring and clear guidelines to ensure ethical and effective use[44,45], our results show the potential for ML-enhanced patient care through the integration of such technologies.

## Discussion

In this work, we propose a swallow detection and clustering approach to allow for a better assessment of LTHRM data. The presented approach automates the assessment of all relevant swallow events in patients with suspected intermittently occurring motility disorders of the esophagus. As a first step, a DL-based swallow detection algorithm is designed to detect all swallow events in raw LTHRM data. It achieves a Recall-score of 94%, surpassing the performance of the only existing commercial software solution, as well as a non-ML baseline. In a second step, the subsequent clustering algorithm is able to cluster the detected swallows into meaningful clusters, which is demonstrated in our evaluation. The combined approach of detecting and clustering the swallow events into groups is resulting in a much simpler representation of the complex data. This simplifies the overall task of analyzing LTHRM data to just a few swallows that need to be evaluated by a medical expert. In our evaluation study, we show that the approach is not only less time-consuming, but the results also indicate that it has the potential to be less stressful for the medical experts. Furthermore, we demonstrate in our evaluation study that the assessment quality remained consistent when using our proposed approach. Consequently, we are able to demonstrate the effectiveness and clinical positive impact of our approach, making LTHRM more feasible and more reliable in clinical care.

The implications of our research are far-reaching. By automating the detection and analysis of swallow events, we reduce the need for extensive manual review by clinicians, which can be time-consuming and subject to human error. This automation not only enhances diagnostic accuracy but also enables healthcare providers to focus more on patient care and less on data processing.

In future steps, we want to improve the UI of our approach to improve the user experience and make the usage of our approach more intuitive, therefore aiming to increase the confidence that users have when using the tool. Furthermore, we plan to improve the proposed algorithm by expanding its application to a broader data set. This expansion aims to integrate not only clustering but also automatic classification of swallows into predefined classes, such as normal and abnormal swallows. This development promises to further streamline the diagnostic process, enabling more precise and efficient identification of swallow characteristics.

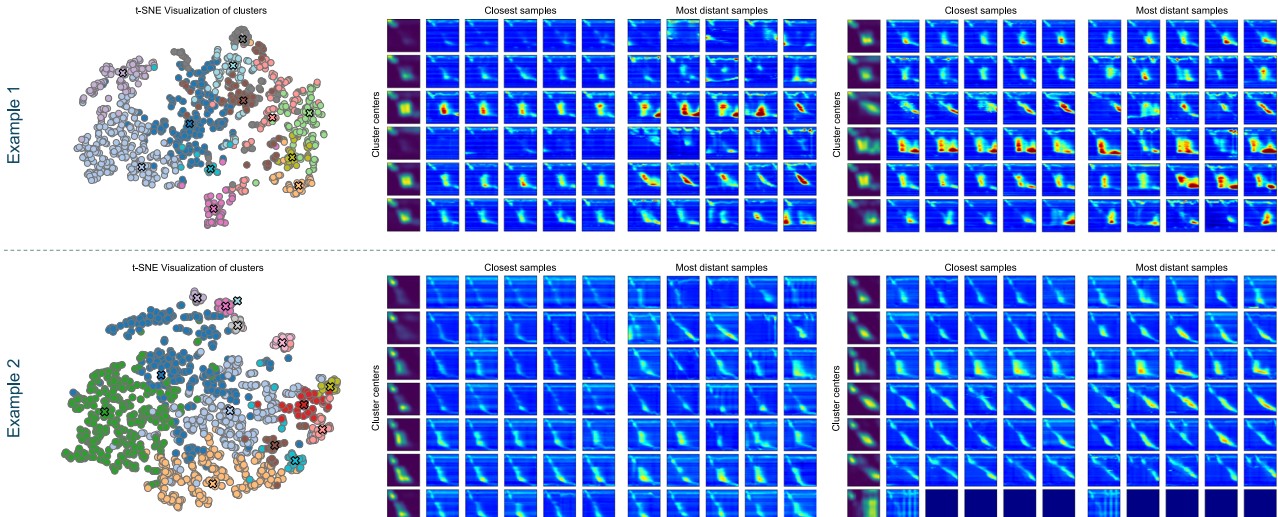

**Fig. 7 | 2D representation of swallow events, closest and most distant samples to their respective cluster center.** For illustration purposes, the clustering results for two patients are shown. The left plots show the representations of all swallows after being reduced to two dimensions using t-SNE, where the swallows are colored according to their assigned cluster. The right plots depict the identified cluster centers and the five closest as well as the five most distant swallows to each center.

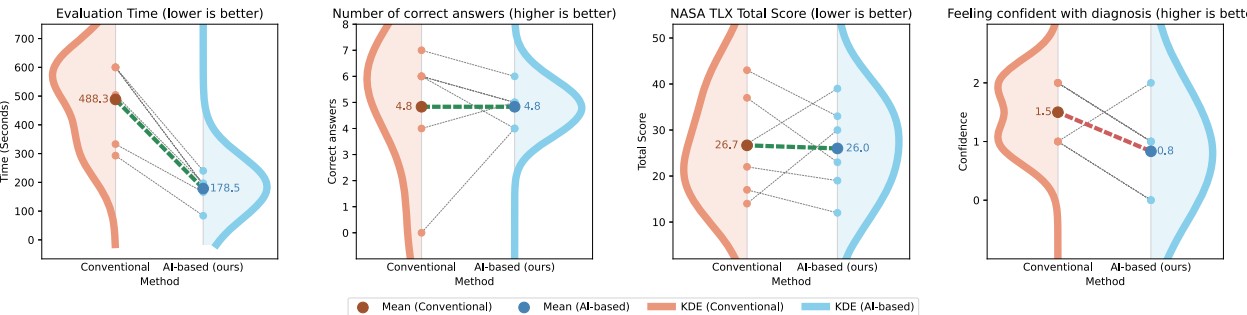

**Fig. 8 | Evaluation study results.** The *Conventional* method is compared with our *AI-based* method in several aspects. The plot displays the paired measurements to show the individual differences, as well as a kernel density estimate (KDE) on each side to give an overview of the overall distribution. When using the *AI-based* method, evaluation time is decreasing, the number of correct answers remains similar, the NASA TLX total score is slightly decreasing, and the confidence of the participants in their diagnosis is decreasing.

Ultimately, this could lead to significant advancements in the treatment of swallowing disorders by providing healthcare professionals with a more nuanced understanding of swallow patterns.

Such advancements could facilitate earlier detection and intervention for patients with esophageal motility disorders, potentially improving patient outcomes and quality of life. Additionally, by leveraging larger and more diverse data sets, we anticipate the enhancement of the algorithm's robustness and generalizability across different patient populations. This will be crucial for its adoption in varied clinical settings and its ability to handle the wide range of presentations associated with esophageal disorders.

Moreover, the integration of more advanced ML techniques and continuous learning systems could further refine the algorithm's performance over time, adapting to new data and emerging trends in patient care. As the healthcare industry increasingly embraces digital health solutions, our approach could serve as a model for other applications, highlighting the potential of AI to revolutionize medical diagnostics and treatment strategies, especially for rare and uncommon diseases.

In conclusion, our swallow detection and clustering method marks a significant step forward in the automated assessment of LTHRM data. Through rigorous testing and validation, we have demonstrated its performance and potential for clinical impact. With future enhancements and broader data integration, we envision a transformative effect on the diagnosis and management of esophageal motility disorders, ultimately contributing to improved patient care and clinical outcomes.

## Data availability

The main data supporting the results of this study are available within the paper and its Supplementary Information. Source data underlying Fig. 8, Tables 1, 2, and 3 can be found in Supplementary Data 1. The data sets generated and analyzed during the current study are not publicly available due to restrictions related to privacy concerns for the research participants, but are available from the corresponding author on reasonable request, subject to approval by the relevant institutional committees and in accordance with all applicable laws and regulations. Requests will be evaluated on a case-by-case basis and processed within 6–8 weeks. Data usage will be governed by a data use agreement that prohibits re-identification and limits usage to the approved scientific purpose.

## Code availability

The underlying code for this study is available at https://github.com/ ResearchgroupMITI/swallow-detection and provided in Zenodo[46]. The following packages were used: Python 3.10.13, PyTorch 2.0.0, Torchvision 0.15.2, Numpy 1.26.0, Pandas 2.1.1, Matplotlib 3.8.0, Scikit-image 0.20.0, Scikit-learn 1.3.0, and Wandb 0.15.12.

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

## Acknowledgements

We acknowledge funding support by AiF Germany for A.G. and A.J. during the data analysis period. No funding was obtained during data collection.

## Author contributions

A.G. and L.W. developed and implemented the proposed methodology. A.J. designed the technical experiments as well as collected and curated the data sets. A.G. and L.W. evaluated all technical experiments. A.G., D.W., and A.J. acquired survey participants and evaluated the survey. D.R. and A.J. defined the clinical goal of the study. A.G. generated the figures for the manuscript. A.G., L.W., and A.J. wrote different parts of the manuscript. All authors reviewed the manuscript.

## Funding

## Competing interests

The authors declare no competing interests.
