## [Transparent Peer Review file · Communications Medicine]

Reviewer #1 (Remarks to the Author):

Comment 1

Please clarify whether data segments recorded during meal times were included in the analysis. If so, provide a rationale for their inclusion. If not, please describe the criteria and methods used to exclude these segments.

Response:

Thank you for the thoughtful question. We included mealtime segments because a representative circadian analysis must capture all phases of daily life (sleep, food intake, activity). To preserve real-world generalizability and avoid biasing time-of-day and activity effects, we analyzed continuous 24-h recordings and applied uniform quality control to remove artifacts, regardless of time of day and activity.

Changes:

In the methods section, we now clarify that the data set included all aspects of daily life, including sleep, food intake, and activity periods. Participants were advised to record their meals, body position, and symptoms while maintaining their daily routines to obtain representative and traceable measurements. We further emphasize that mealtime segments were not excluded, as their inclusion was essential for capturing realistic circadian and behavioral patterns.

Comment 2

The authors use data from all 36 sensors to detect swallows, but they do not explicitly account for the time differences in signal propagation across the sensors. A swallow event initiates in the pharynx and progresses down the esophagus. Therefore, the pressure changes associated with the swallow will be detected by the proximal sensors before the distal sensors. The authors do not specify which sensor's signal or what method they used to determine the single "swallow start" time used for labeling and windowing. Besides there is high inter observer variability in determining start of swallows. how were the labels provided? Please Specify the criteria.

Response:

Thank you for highlighting this important point. In our dataset, the swallow start is defined by the relaxation of the upper esophageal sphincter (UES), which reliably marks the initiation of a swallow event. For training, swallow windows are aligned such that the UES relaxation corresponds to the left edge of the analysis window. This approach ensured that the temporal progression of the swallow through proximal to distal sensors was preserved within the window. During inference, we assume that the detection confidence is maximized when the swallow start aligns with the left edge of the analysis window. Therefore, we look for the largest value in s across the sequence and use that index as the predicted start. (See section "Methods" and Figure 3 for the processing details).

To minimize inter-observer variability in labeling, all swallow events were first annotated by a single experienced medical expert. These initial annotations were then independently reviewed and verified by a second expert to ensure consistency and accuracy.

Changes:

Considering your valuable feedback, we now clarify in the methods section that the swallow start was defined by relaxation of the upper esophageal sphincter. We also now specify how the predicted start time was determined during model inference, namely by identifying the time point with the maximum detection confidence, and elaborate the rationale behind it. Furthermore, we include a statement about our labeling strategy to ensure high-quality labels.

Comment 3

While the rescaling of the input matrix I is understandable for standardization, it might inadvertently diminish the importance of subtle pressure variations critical to manometric interpretation. please justify how the rescaling of the input matrix I preserves critical spatio-temporal information. Have the authors considered using a trainable filter (e.g., a convolutional layer) to learn relevant features from the original, unscaled data, potentially avoiding the information loss associated with a global rescaling operation?

Response:

Thank you for this thoughtful remark. The rescaling step in our preprocessing pipeline serves two purposes: (i) standardization of input values across subjects and recordings, and (ii) clipping of extreme outliers that fall outside the physiologically valid pressure range. Importantly, this operation does not distort the relative variations in pressure across time or sensors, but only maps them into a normalized range. Thus, the critical spatio-temporal patterns underlying the swallow detection are preserved.

We acknowledge that alternative approaches, such as incorporating a trainable filter (e.g., convolutional layers) directly on the raw, unscaled data, are an interesting future direction. However, in this work we intentionally adopted a non-trainable preprocessing strategy to reduce methodological complexity and to isolate the contribution of the swallow detection framework itself.

Changes:

We now provide a more detailed description of the preprocessing procedure in the methods section.

Comment 4

According to figure 5, the authors may ask the participants to provide a score to determine which fraction of these special cases were really attributed to a disorder and also which fraction of the main swallows were not attributed to a disorder.

For future work, I suggest the authors focus on improving the detection and classification of 'special cases'. When evaluating the model's performance, prioritize reporting metrics specifically for these less frequent but clinically important events, as the current model may be biased towards the more common 'main swallows'.

Response:

We appreciate this important suggestion for future work. In our current study, the threshold (15% of all samples) used to differentiate between main swallows and special swallows was already defined in close collaboration with a medical expert. This ensured that the criterion itself was chosen specifically to best separate the main cases from the more special cases in a medically meaningful way. In that sense, the distinction between main and special cases is already implicitly embedded in our current evaluation. Nevertheless, we agree that future work should place greater emphasis on refining the detection and classification of these less frequent but clinically significant events, including reporting dedicated performance metrics for them.

Changes:

According to your comment, we add a clarification in the methods section explaining that the threshold distinguishing main from special swallows was defined together with a medical expert to ensure clinical relevance.

Comment 5

On page 7, for the swallow detection results, please include a holdout validation method specifically applied to the two "special cases" introduced in Figure 6.

Response:

Thank you for the remark. Regarding the two cases shown in Figure 6, both cases correspond to two individual patients who were also part of the overall data set and evaluation cohort. Consistent with our general evaluation protocol to not train the model on the patients that are evaluated, the swallow detection model was not trained on either of these patients. Instead, as you suggest, we already included both cases in the holdout validation set during training, ensuring that the evaluation for these examples was fully independent of the training process.

Comment 6

On Page 8, the manuscript references "Figure 9," but the figure numbering appears inconsistent. Please verify and correct figure references to maintain numerical order, possibly by refreshing the citation management tool.

Response:

Thank you for this remark, this happened because the referenced Figure is part of the supplementary material.

Changes:

We have now separated the supplementary material into a separate document to ensure a more consistent numbering in the main manuscript.

Reviewer #2 (Remarks to the Author):

Comment 1

"disorders of esophageal motor function, such as dysphagia" Dysphagia is not a disorder of the esophagus alone but may involve the oral and pharyngeal swallowing mechanism

Response:

Thank you for this helpful clarification. We agree that dysphagia is not limited to esophageal motor disorders, as it may also involve oral and pharyngeal swallowing mechanisms.

Changes:

To avoid potential ambiguity, we have revised the text in the Introduction accordingly.

Comment 2

Table 2 is not referred to in the text

Response:

Thank you for pointing this out.

Changes:

We now explicitly refer to Table 2 in the results section and provide a brief interpretation of the comparison between different backbone models.

Comment 3

Numbering of Figures is off. Figure 9 is mentioned in the text before Figures 7 and 8

Response:

Thank you for this remark, this happened because the referenced Figure is part of the supplementary material.

Changes:

We have now separated the supplementary material into a separate document to ensure a more consistent numbering in the main manuscript.